# Validity of the Espiro Mobile Application in the Interpretation of Spirometric Patterns: An App Accuracy Study

**DOI:** 10.3390/diagnostics14010029

**Published:** 2023-12-22

**Authors:** Darinka Savic-Pesic, Nuria Chamorro, Vanesa Lopez-Rodriguez, Jordi Daniel-Diez, Anna Torres Creixenti, Mohamed Issam El Mesnaoui, Viviana Katherine Benavides Navas, Jose David Castellanos Cotte, Iván Abellan Cano, Fátima Alexandra Da Costa Azevedo, María Trenza Peñas, Iñaki Voelcker-Sala, Felipe Villalobos, Eva-María Satue-Gracia, Francisco Martin-Lujan

**Affiliations:** 1Camp de Tarragona Primary Care Unit, Institut Català de la Salut, Doctor Mallafrè Guasch, 4, 43005 Tarragona, Spain; dsavic.hj23.ics@gencat.cat (D.S.-P.); esatue.tgn.ics@gencat.cat (E.-M.S.-G.); 2ISAC Research Group, Fundació Institut Universitari per a la Recerca a l’Atenció Primària de Salut IDIAP Jordi Gol, Gran Vía de Les Corts Catalanes, 591 Ático, 08007 Barcelona, Spain; fvillalobos@idiapjgol.info; 3School of Medicine and Health Sciences, Universitat Rovira i Virgili, Carrer de Sant Llorenç, 21, 43201 Reus, Spain; 4Pneumology Service, Hospital Universitari de Tarragona Joan XXII, Institut Català de la Salut, Doctor Mallafrè Guasch, 4, 43005 Tarragona, Spain; 5Primary Care Unit, Sanitat Conselleria, Generalitat Valenciana, Dpto 18, Carretera de Sax s/n, 03600 Elda, Spain; 6Health Centre Group Cávado I, Largo Paulo Orósio, 4700-036 Braga, Portugal; 7Centro de Salud Aguilas Sur, Primary Care Unit, Servicio Murciano de Salud, Calle Rey Carlos III, s/n, 30880 Aguilas, Spain; 8College of Medicine and Public Health, Flinders University, Flinders Drive, Bedford Park, SA 5042, Australia; 9Primary Care Research Support Unit Reus-Tarragona, Institut Català de la Salut, Camí de Riudoms, 53–55, 43202 Reus, Spain

**Keywords:** spirometry interpretation, clinician tools, primary care, respiratory illness

## Abstract

Spirometry is a pulmonary function test where correct interpretation of the results is crucial for accurate diagnosis of disease. There are online tools to assist in the interpretation of spirometry results; however, as yet none are validated. We evaluated the interpretation accuracy of the Espiro app using pulmonologist interpretations as the gold standard. This is an observational descriptive study in which 118 spirometry results were interpreted by the Espiro app, two pulmonologists, two primary care physicians, and two residents of a primary care training program. We determined the interpretation accuracy of the Espiro app and the concordance of the pattern and severity interpretation between the Espiro app and each of the observers using Cohen’s kappa coefficient (*k*). We obtained a sensitivity and specificity for the Espiro app of 97.5% (95% confidence interval (CI): 86.8–99.9%) and 94.9% (95%CI: 87.4–98.6%) with pulmonologist 1 and 100% (95%CI: 91.6–100%) and 98.7% (95%CI: 92.9–99.9%) with pulmonologist 2. The concordance for the pattern interpretation was greater than *k* 0.907, representing almost perfect agreement. The concordance of the severity interpretation was greater than *k* 0.807, representing substantial to almost perfect agreement. We concluded that the Espiro app is a valid tool for spirometry interpretation.

## 1. Introduction

Pulmonary function tests (PFTs) are a range of clinical tools to evaluate and monitor the progress of patients with respiratory symptoms like cough, dyspnea, or established lung disease [1]. In primary care, PFTs are used to confirm the diagnosis of respiratory diseases and assess the impact of therapeutic interventions on pulmonary function [2].

Some PFTs, like spirometry and pulse oximetry, are widely used; others, such as lung diffusion capacity or lung volume tests, are performed in specialized respiratory laboratories. Spirometry is the most widely used PFT in primary care and is considered the gold standard to evaluate the most common causes of pulmonary disease by international organizations [1]. Guidelines recommend the use of spirometry to monitor the progression and severity of respiratory diseases such as asthma and chronic obstructive pulmonary disease (COPD) [3,4].

Primary care plays an important role in the early diagnosis of respiratory diseases. Although spirometry is widely used, there are still many issues with obtaining reliable data and interpreting the results to provide the best support for clinical decisions [5]. It has been observed that clinicians are relatively inaccurate at predicting both the severity of airflow limitation and the nature of that limitation, leading to under- and over-diagnosis of pulmonary diseases. Many barriers delay diagnosis in primary care, including underreporting of symptoms by patients, limited access to spirometry, a lack of recognized and effective case-finding methods, and a lack of expertise in spirometry interpretation [6,7].

Artificial intelligence-powered medical technologies are rapidly evolving; these tools enable health professionals to provide patients and society with better quality of life through early diagnosis, thereby reducing complications, optimizing treatment, providing less invasive options, and reducing the length of hospitalization [8]. In recent years, mobile health apps have made healthcare more accessible and affordable for everyone. However, the number of apps has grown exponentially, with almost no control or regulation of any kind, and very few of them have undergone a validation process, which reduces confidence in them among health professionals [9]. In pulmonary medicine, the interpretation of PFTs has been reported as a promising field for the development of applications [8]. In a recent study, the artificial intelligence-based software perfectly matched the PFT pattern interpretations (100%). In contrast, pulmonologists’ pattern recognition of PFTs matched the guidelines by 74.4% [10].

In 2016, the Espiro app was created in Catalonia by the members of the Catalan Society of Family and Community Medicine (CAMFIC). It was developed to evaluate lung function from spirometry data, automatically interpreting the results and proposing a preliminary diagnosis. This app makes it easier for healthcare professionals to interpret spirometry results but is yet to be validated. We expect validation to increase the use and confidence in the app among health professionals and health institutions.

Therefore, the aim of this study was to assess the interpretation accuracy of the Espiro app using the pulmonologist interpretation as the gold standard and to summarize the correlation coefficients between interpretations obtained from the Espiro app and pulmonologists, primary care physicians, and residents of a primary care training program.

## 2. Materials and Methods

### 2.1. Study Design

An observational descriptive study was carried out for two consecutive years (2019–2020), with the participation of two pulmonologists, two primary care physicians, and two primary care physicians in a training program (last year of the program). They independently evaluated and interpreted complete spirometry reports (pre- and post-bronchodilator results, with graphics). The interpretation was performed without following a pre-established protocol; we recommended that they follow their usual clinical practice. The same spirometry reports were interpreted using the Espiro app.

### 2.2. Study Sample

The study included spirometries obtained from the MEDISTAR study (Mediterranean diet and smoking in Tarragona and Reus), a controlled clinical trial that evaluated the effectiveness of an educational intervention to increase Mediterranean diet adherence in a sample of smokers with no previous respiratory disease [11].

We selected 118 spirometries by convenience sampling to secure a minimum number of common spirometric patterns (normal, obstructive, spirometric restriction, and mixed) with different grades of severity. All the selected spirometries were performed with standardized equipment by respiratory technicians (spirometer model DATOSPIR-600 with a disposable Lillytype transducer; SIBELMED, Barcelona, Spain) and fulfilled the quality criteria defined by the Spanish Society of Pulmonology and Thoracic Surgery (SEPAR) [12]. We included 45 impaired spirometries (equivalent to “sick”), anticipating that the Espiro app would correctly detect (have a sensitivity of) 90% or more; this would allow for precision in the 95% confidence interval of ±7%.

### 2.3. Espiro App

Members of the CAMFIC respiratory disease group developed the software, which has been available for the Android (Play Store) and iOS (App Store) operating systems since February 2016 (the version currently available is 1.6). It supports five different languages (Catalan, Spanish, English, French, and Portuguese). At first, the app asks for anthropometric data (sex, age, height, and weight), as well as the selection of a reference table from GLI-2012 (Global Lung Function Initiative) that covers a wide age range in different ethnic groups [13]. Secondly, the spirometric data (forced vital capacity (FVC) and forced expiratory value in the first second (FEV1)) obtained in basal and postbronchodilator spirometry must be introduced. Finally, the Espiro app generates a report based on a comparison of the theoretical values and the anthropometric data using a previously validated algorithm adapted to the American Thoracic Society/European Respiratory Society (ATS/ERS) spirometry recommendations [14]. The report indicates the functional pattern (normal, obstructive, spirometric restriction, or mixed), severity (mild, moderate, moderately severe, severe, and very severe), and the result of the reversibility test, commonly called the bronchodilation test (negative without changes, significant reversibility, and non-significant reversibility).

The app does not evaluate the technical acceptability of the spirometric technique because it was designed to interpret the spirometric patterns. The evaluation of the quality of the technique needs to be carried out by the technician who performs the test or the physician who evaluates it.

The spirometry measures the maximal volume of air that can be breathed as a function of time. The most relevant spirometric measurements include the total volume from a maximally forced expiratory effort after the deepest inhalation (FVC); the volume of expiratory air in the first second of the expiration (FEV1), and the relationship between the two (FEV1/FVC). Through systematic interpretation, physicians can identify the different patterns of impaired airflow associated with several pulmonary diseases. To evaluate the reversibility pattern, spirometry is performed after the administration of a short-acting bronchodilator [1]. Although spirometry contains many other parameters of respiratory flow, they are not usually employed in the interpretation.

According to the current recommendations, we were able to classify the results based on the following spirometric criteria:Pattern: The results of the spirometry can be divided into two large classes: normal and impaired. The impaired results can be classified into three disorders: obstructive, spirometric restriction, and mixed [12]. This allows the obtained results to be sorted into 4 patterns:
Normal: an FVC% ≥ 80% of the predicted value, an FEV1% ≥ 80% of the predicted value, and an FEV1/FVC ratio ≥0.7 in absolute value.Obstructive: an FVC% > 80% of the predicted value and an FEV1/FVC ratio < 0.7 in absolute value.Spirometric restriction: an FVC% < 80% of the predicted value and an FEV1/FVC ratio ≥ 0.7 in absolute value.Mixed: an FVC% < 80% of the predicted value, an FEV1% < 80% of the predicted value, and an FEV1/FVC ratio <0.7 in absolute value.

At this point, we highlight that the use of a fixed ratio such as 70% for FEV1/FVC is controversial because it is known that the normal value (lower limit of normal (LLN)) is age-dependent, and this results in overestimation of the presence of obstruction in older patients and underestimation in younger patients [15].

2.Severity of airflow limitation: quantified by the degree of reduction in FVC or FEV1 value (expressed as a percentage of the predicted normal) as mild (>70%), moderate (60–69%), moderately severe (50–59%), severe (35–49%), and very severe (<35%) [12].3.Bronchodilator test: A bronchodilator, such as short-acting beta2-agonists, is often administered during spirometry so that airway responsiveness can be assessed. The degree of bronchodilator response may be expressed as a percentage increase or absolute increase (in milliliters), or both, compared with the pre-bronchodilator value. It is considered positive if there is an improvement in the FEV1 or the FVC ≥ 12% and ≥0.2 L, not significant if there is an improvement <12%, and negative if not reverted or not modified [12].

Figure 1 shows the different screens that appear in the Espiro app.

### 2.4. Procedure

The selected spirometries were distributed to the different professionals through packages of printed reports, with each one showing age, sex, anthropometric data (weight, height, and body mass index), spirometric values grouped into five columns (reference value, obtained basal value, obtained/reference percentage, post-bronchodilator value, percentage of basal increase/post-bronchodilator), and two graphics (volume–time curve and flow–volume curve). Any identifying data from the patient were masked with a dual purpose: (1) to comply with data protection and (2) to hide any element that allows observers to identify the reports.

For the selection of the participants, it was considered whether any of them had any type of relationship with the app developers or investigation collaborators to avoid any type of bias in the interpretation of the spirometries.

The interpretation of the spirometries was carried out in three blocks. The blocks consisted of 60 set spirometries. The spirometric patterns were evenly distributed, with a minimum proportion of 10% for each of the impaired patterns (obstructive, spirometric restriction, and mixed). A control number of repeated spirometries from the first block, blinded to the participant professionals, was included in the second and third blocks to assess intra-rater reliability.

The professionals’ interpretations were performed as in their daily practice, without a specific form or instructions. As soon as they finished the analyses of a block, they were provided with the next, with each block released on a monthly basis.

### 2.5. Study Variables

The main variable of interest was the spirometric pattern detected by the app and the observers. As secondary variables, we also evaluated the disease severity and the bronchodilator test results.

According to ATS/ERS recommendations [16], we categorized the spirometric patterns into normal, obstructive, spirometric restriction, and mixed. In addition, we categorized severity into mild, moderate, moderately severe, severe, and very severe. Finally, we categorized the bronchodilator test into positive, negative, and not modified.

### 2.6. Statistical Analysis

All the results were transferred to a database by the same observer to avoid bias. To calculate the diagnostic accuracy, we considered the pulmonologists’ interpretations the gold standard, and the pattern variable was converted into a dichotomous variable (normal and impaired), considering “sick” patients to be those with impaired spirometry and “healthy” patients to be those with normal spirometry. Regarding the app interpretation, the obstructive, spirometric restriction, and mixed patterns were considered positive and the normal pattern negative. The results were plotted in 2-by-2 tables and the sensitivity, specificity, positive predictive value (PPV), negative predictive value (NPV), positive likelihood ratio (+LR), negative likelihood ratio (−LR), and overall accuracy (with 95% confidence intervals (CI)) were calculated using the website MedCalc [17] for each of the pulmonologist observers.

The concordance of the pattern and severity interpretation between the Espiro app and each of the observers (2 pulmonologists, 2 primary care physicians, and 2 primary care physician training program residents), were analyzed by determining the Cohen’s kappa coefficient (*k*) (with 95% CI). Severity data were pooled into three groups: (1) mild, (2) moderate to moderately severe, and (3) severe to very severe. The *k* results were interpreted as follows: Values ≤ 0 indicate no agreement, 0.01–0.20 indicate none to slight, 0.21–0.40 indicate fair, 0.41–0.60 indicat4e moderate, 0.61–0.80 indicate substantial, and 0.81–1.00 indicate almost perfect agreement [18].

Analyses and data handling were performed using the R Statistics package (R Foundation for Statistical Computing, Vienna, Austria; version 4.0.5).

## 3. Results

We initially calculated the validity of the test with a total study sample of 118 spirometry reports.

Table 1 presents the cross-tabulation of spirometry pattern interpretation (normal and impaired) by the Espiro app and each pulmonologist. The percentage of impaired spirometries in the selected sample was 34% for pulmonologist 1 and 36% for pulmonologist 2.

Table 2 lists the app accuracy values compared with the two pulmonologists used as the gold standard. Overall, the results obtained from the two pulmonologists were similar. We highlight that the Espiro app had a high level of accuracy with both pulmonologists, with a sensitivity and specificity of 97.5% (95% confidence interval (CI): 86.8–99.9%) and 94.9% (95%CI: 87.4–98.6%) with pulmonologist 1 and of 100% (95%CI: 91.6–100%) and 98.7% (95%CI: 92.9–99.9%) with pulmonologist 2. In addition, the ROC analysis indicated that the area under the curve (AUC) was close to 1 with both pulmonologists (Figure 2). There are no significant differences between the two pulmonologists’ results.

Table 3 shows the correlation between the Espiro app pattern interpretations and each pulmonologist, differentiating between normal, obstructive, spirometric restriction, and mixed patterns. In the case of pulmonologist 1, we joined the obstructive and mixed patterns of the app interpretation to complete the cross-tabulation, because the pulmonologist did not differentiate between these two patterns and included the mixed pattern with the obstructive one. The correlation percentages of the different patterns interpreted by the Espiro app and each pulmonologist were higher than 90% for the normal, obstructive, and spirometric restriction patterns and somewhat less in the mixed pattern for pulmonologist 2 (81.3%). The kappa agreement for the Espiro app and the pulmonologists was 0.885 for pulmonologist 1, and 0.923 for pulmonologist 2.

Table 4 shows the concordance for pattern interpretation (normal or impaired) of the Espiro app and each of the observers. The concordance results obtained for the pattern interpretation between the Espiro app and each of the observers were greater than 0.90. The intra-rater reliability was greater than 0.920 for all observers. All the results were statistically significant, with *p* < 0.001.

Finally, Table 5 shows the concordance for severity interpretation (mild, moderate/moderately severe, and severe/very severe) of the Espiro app and each of the observers. The *k* concordance was greater than 0.877, except for pulmonologist 1, for whom k was 0.807. The intra-rater reliability was greater than 0.843, except for one of the primary care physician training program residents, for whom *k* was 0.784. It is noteworthy that the concordance obtained between pulmonologist 1 and the other observers was substantial (*k* ≥ 0.716), unlike the concordance obtained among the other observers. All the results obtained were statistically significant, with *p* < 0.001.

The concordance for the bronchodilator test could not be calculated due to insufficient interpretations performed by the different observers.

## 4. Discussion

In the current study, we explored the accuracy of the Espiro app in the interpretation of the spirometry results. We obtained an app accuracy greater than 95% compared with two pulmonologists who are experts at evaluating respiratory function tests, along with an almost perfect inter-rater reliability in pattern interpretation and very good reliability for severity interpretation (*k* > 0.90 and *k* > 0.80, respectively).

Spirometry is the most commonly used pulmonary function test in the diagnosis and monitoring of patients with pulmonary diseases in primary care centers [19]. In clinical practice, its interpretation is based on the detection of patterns of functional impairment [2]. However, these patterns are not always obvious or easily recognizable to professionals unaccustomed to interpreting them [20]. To improve clinical practice, a mobile software tool was designed to assist in the interpretation of spirometry. It incorporates a decision-making algorithm based on recently updated international guidelines for spirometry interpretation [1]. Our initial intention was to aid interpretation using the Espiro app to increase the reliability of primary care physicians, as well as optimize and reduce the time invested in spirometry interpretation. The results of this study may support this.

Currently, it is undisputed that access to spirometry in primary care is an unavoidable necessity for the care of patients with respiratory diseases such as asthma or COPD [21]. Although the benefits of spirometry as a population-screening tool are uncertain, the evidence does not contradict its use to diagnose patients with risk factors (mainly smoking) and/or symptoms suggestive of respiratory disease [22]. Despite spirometry being a relatively simple technique, periodic training is necessary to acquire and maintain skills [1,2]. Even when assuming that the test methodology is correct, questions remain about the reliability of the data and how they should be interpreted to better support clinical decision-making [23]. In addition to proper training, the correct interpretation of spirometric tests requires time and regular performance of a sufficient number of spirometric tests in order to maintain the required skills [24]. In our setting, family physicians interpret most of the spirometry tests performed in primary care, but only some of them acknowledge having received specific training [25]. Thus, facilitating clinical practice with computer-based decision support tools could be an alternative. This is not a new idea, and such systems have been shown to improve the performance of physicians [26].

Diagnosis of disease using automated algorithms holds enormous potential [27]. Many tools are already in development or use, and the potential for their effective integration into routine care settings has never been stronger [28,29]. Spirometry is an ideal medical test for the development of automated interpretation algorithms [30], as the practice and interpretation of this test are well standardized and accepted worldwide [1]. Indeed, several algorithms exist to facilitate the interpretation of spirometry in clinical practice. However, there is limited information in the literature describing how useful they can be in guiding physicians while interpreting spirometry data.

In this study, we assessed the efficacy of the Espiro app for this purpose. It has previously been evaluated by entities for the development of social e-health projects, ranking among the best [31]. It has had more than 11,500 downloads for both Android (Play Store) and iOS (AppStore) and is regularly used by more than 6000 healthcare professionals. It is based on a previously validated algorithm that follows the SEPAR recommendations for the interpretation of spirometry [12,14]. This does not in itself guarantee that the obtained interpretation is correct. A recent review concluded that there is considerable variability among the tools available as diagnostic aids and that some do not comply with guideline recommendations for diagnosis, whereas others were considered impractical for use in primary care [20]. Perhaps for this reason, they are not commonly used in daily clinical practice despite having been available for years [32].

Inter-observer variability in the interpretation of medical tests is well recognized. In PFTs, the literature indicates sizeable room for improvement in the accuracy of results that physicians provide, because not all of them are using the same interpretative strategies in their daily routines [33]. To evaluate the Espiro app’s accuracy, we compared it with the interpretation of two pulmonologists as the gold standard. The accuracy in detecting impaired tests was very high and similar between the Espiro app and the two pulmonologists (95.8% (95%CI 90.4–98.6%) and 99.2% (95%CI: 95.4–99.9%), respectively. In addition, the ROC analysis yielded AUCs of 0.947 and 1, representing high sensitivity and specificity in pattern interpretation by the Espiro app. Regarding the Espiro app’s pattern interpretation, differentiating by normal, obstructive, spirometric restriction, and mixed, with each pulmonologist the kappa agreement can be considered almost perfect (0.885 and 0.923 for pulmonologists 1 and 2, respectively). A recent meta-analysis found that deep learning algorithms have equivalent accuracy to that of healthcare professionals [34]. Although this estimate seems to support the claim that deep learning algorithms can match clinician-level accuracy, it should be considered that clinical decision support systems are not perfect instruments and will have failures [27]. Despite this, we consider that the results represent an almost perfect agreement between the Espiro app and the pulmonologists, as they fall within the range reached by some expert clinical panels and probably have little margin for improvement [35].

In the literature, we found studies that evaluated different software for the interpretation of spirometries, but none of them was a mobile app. In 2013, Nandakumar et al. evaluated a software for spirometry interpretation and found an average accuracy of 95.74% [36]. More recently, in 2022, Wang et al. explored the accuracy of deep learning-based analytic models based on flow-volume curves; they found that one of the models exhibited an accuracy of 95.6% when interpreting ventilatory patterns and that the physicians had an accuracy of 76.9 ± 18.4% [37].

In our study, when we analyzed the pattern interpretations by two primary care physicians and the two residents in a primary care training program, we found an almost perfect correlation with the Espiro app, which makes us think that they used similar techniques for the interpretation of the spirometry results. Although it is unclear whether an automated algorithm has any added value to expert physicians, it could be helpful to less experienced professionals. Moreover, our findings are particularly relevant in primary care, as doctors are very likely to use diagnostic help support tools if they can obtain meaningful and clear information quickly. Rapid access to accurate and relevant information could change clinical outcomes and benefit common clinical practice [38].

In the present study, we observed a small difference between the two pulmonologists (concordance coefficient kappa for pattern interpretation 0.888 (95%IC 0.799–0.976)). Although one of the pulmonologists did not differentiate between the obstructive and mixed pattern and included them both in the obstructive one, the other pulmonologist did differentiate them. In both cases, the agreement was almost perfect when we compared the interpretation of the different patterns. This could be because some pulmonologists differentiated only between obstructive and non-obstructive spirometric patterns. In fact, although all of them agree on the need to conduct other tests to confirm a spirometric restriction, in this study only the spirometric pattern interpretation was evaluated, without a final diagnosis. This is not exceptional, as marked variations in the interpretation of PFTs by pulmonologists has been noted. In a study carried out by Topalovic et al. in 2019 [10], they observed a considerable inter-rater variability between 118 pulmonologists since the interpretations of the pattern only matched 75% of the cases (*k* 0.67), which demonstrates that the task is fundamentally prone to disagreements. In contrast, an app does not depend on individual interpretation; rather, it is a system based on an algorithm that consistently analyzes the data in the exact same manner.

Staging pulmonary disorder severity using PFTs is an invaluable guide for the prognosis of respiratory diseases [39]. This study also showed high inter-rater reliability in the severity interpretation, ranging from substantial (*k* 0.807) to almost perfect (*k* 0.956). Comparing the results of both pulmonologists, it is remarkable that the quality of the classification differed only slightly, contrary to other studies that showed greater variation [10]. The differences and the lack of severity interpretation in some cases by the observers may be due to the different interpretation algorithms used. A recent review by D’Urzo et al. found considerable variability among spirometry interpretation algorithms and a need for standardization in primary care [20]. In the case of the Espiro app, the interpretation never failed because it follows the established parameters and always interprets the data using the same algorithm.

### 4.1. Limitations and Strengths

Computerized clinical decision support systems for diagnosis can generally be built on linkages between clinical data and gold standards for accuracy. Its scientific foundation must be strong, with rigorous and updated evidence, establishing its validity, usability, and reliability. Furthermore, they should provide evidence that they have implemented processes to ensure that the system maintains a current knowledge base and that it is safe to use [27]. The Espiro app has been designed following internationally accepted interpretation standards and is updated according to the latest recommendations of clinical guidelines, and for its validation, we use as a gold standard the reports of two recognized expert pulmonologists in the same task.

Despite the good results that deep learning-based analytic models have shown in diagnostic performance and decision-making support in clinical environments, this enthusiasm should not override the need for critical evaluation. Concerns raised include whether the findings are generalizable and to what extent the results are applicable to a real-world clinical setting [34]. Likewise, the current study has some limitations that need to be commented on. The design of the study was to evaluate the app’s accuracy in pattern recognition alone; as such, it does not represent real clinical practice where tests are interpreted with the patient context in mind. Although the definitions of the spirometric patterns do not take any other clinical data into account, in clinical practice a physician can choose whether to use additional information beyond that provided by PFTs before making the diagnosis. The PFTs, together with the patient’s symptomatology, can serve as a basis for differentiating between spirometric restriction, obstructive, and mixed causes of lung disease [19]. Furthermore, in this context, the interpretation of a “normal” spirometry report could lead to a false sense of security among doctors who may consider this finding an absence of sickness. Therefore, we recognize that considering the deep learning algorithms in this isolated way can limit the capacity to extrapolate the findings in this study to standard clinical practice.

In addition to the baseline data, all spirometry tests assessed in this study included the bronchodilator test, regardless of whether the FEV1/FVC was less than 0.70 or normal. In fact, it is known that a normal value of FEV1/FVC is not synonymous with the absence of elevated airway tone; as such, administering a bronchodilator drug such as a β2-agonist may still result in clinically relevant improvements [40]. Consequently, bronchodilator test can still have important clinical implications [41]. For example, a major response to a bronchodilator can suggest an asthma diagnosis, whereas a persistent or poorly modifiable obstruction with a bronchodilator in a smoking patient suggests COPD [39,42]. Although bronchodilator test data were available to observers in this study, concordance could not be calculated due to a lack of test interpretation by some observers.

In our study, we reported the formal sample size calculation to ensure that the study was of sufficient size in direct comparison of altered patterns between the Espiro app and the pulmonologists. We want to highlight that this calculation is not always conducted in other studies that compare automatic methods with physician’s reports because of the lack of consensus about the methods based on the principles to conduct them [34]. However, the sample we used may not have been large enough to detect differences in the interpretations between the pulmonologists and the app, as it was taken from a population of previously healthy smoking patients [11]. Although we believe it reflects the prevalence of patterns faced by family physicians in their daily clinical practice, one might think that a study based on a sample with a higher prevalence of altered or more-difficult-to-interpret cases might have rendered different results [43]. Moreover, internal validation overestimates diagnostic accuracy for healthcare professionals and deep learning algorithms. Therefore, these findings highlight the need for out-of-sample external validation in all predictive models [34].

### 4.2. Implications for Clinical Practice and Research

Overall, the results of the study showed high accuracy in the interpretation of spirometry by the Espiro app. We therefore consider that the Espiro app is an effective resource for answering the different spirometry questions that physicians must pose, making it a valid tool to help family physicians. Therefore, we can extrapolate that introducing this kind of software to a primary care system could reduce diagnostic errors and optimize time management for health professionals. In addition, this can be a tool for family physicians to use as an aid during their training program. However, the information delivery must be tailored to the experience level of the user, making it clear that it is designed to assist and inform but not replace clinical reasoning. On the other hand, we should note that patient and technical factors may affect the quality of the data for interpretation by an algorithm. The spirometries must be performed by a trained technician because the software cannot identify the quality of the data. In our setting, clinical spirometry results are deemed reliable and accurate when certified, calibrated devices are used by trained spirometry operators in accordance with standard criteria and when spirometry maneuvers are performed on appropriately prepared patients [1]. To enable a comprehensive utilization of the application, we emphasize the importance of evaluating the quality of spirometry. This necessitates the training of both the technician conducting the test and the doctor interpreting the results. Additionally, it is noteworthy that there is artificial intelligence-based software designed for assessing the technical quality of the spirometric test [44].

Clinicians and researchers have long envisioned that computer tools could assist decision-making in clinical practice [27]. The impact of assisted tools could change according to the context. Intuitively, in the clinical setting of primary care physicians who are less habituated to spirometry interpretation, the use of a tool such as the Espiro app could increase the precision in interpreting the disease pattern and severity levels from spirometry data, thereby improving diagnosis and helping to make better treatment decisions. In any case, the potential benefit and the risks related to excessive confidence in the App need to be evaluated in follow-up studies.

## 5. Conclusions

In conclusion, our results indicate that the Espiro app, an accessible and free mobile application, provides spirometry result reports equivalent to the interpretations made by pulmonologists. This app is a valid tool that can be used to support family physicians’ decision-making and improve clinical practice in primary care.

## Figures and Tables

**Figure 1 diagnostics-14-00029-f001:**
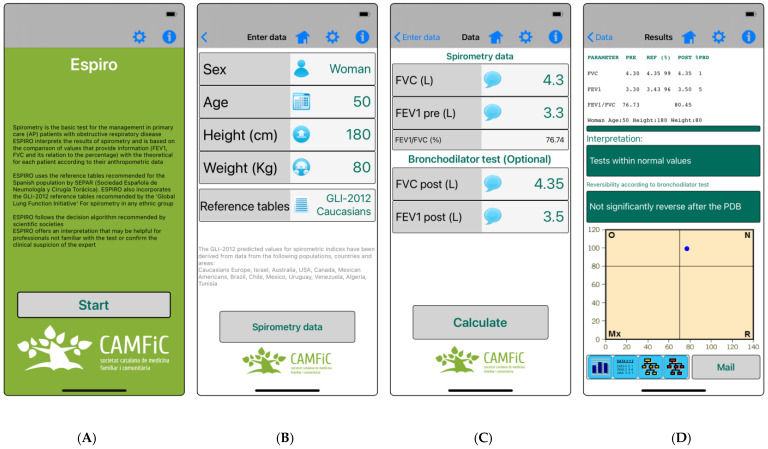
Different screens of the Espiro app. (**A**) initial screen; (**B**) patient data introduction and reference table choice (**C**); spirometric data and bronchodilator test data introduction; (**D**) interpretation of the pattern and severity (upper box), bronchodilator test (lower box), and graphic representation of the spirometric pattern (blue circle).

**Figure 2 diagnostics-14-00029-f002:**
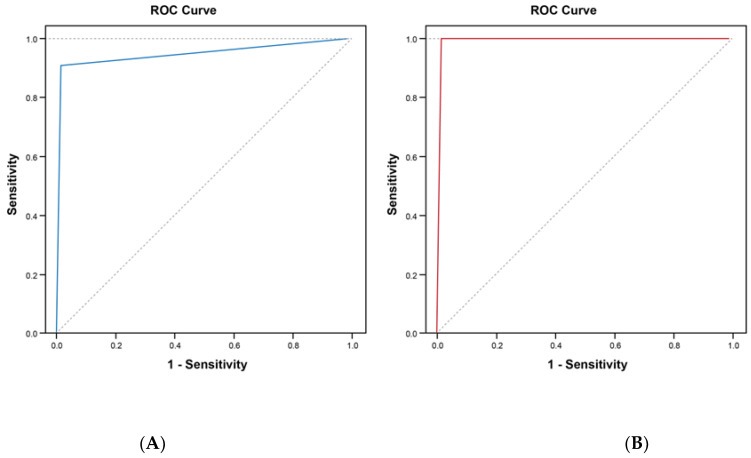
ROC (receiver operating characteristic) curve of the Espiro app to predict the diagnosis of each pulmonologist. (**A**) Area under the curve for pulmonologist 1: 0.947 (95% confidence interval: 0.894–1.000); (**B**) area under the curve for pulmonologist 2: 1.000.

**Table 1 diagnostics-14-00029-t001:** Cross-tabulation of spirometry pattern interpretation (normal and impaired) by the Espiro app and each pulmonologist.

		Pulmonologist 1 Interpretation			Pulmonologist 2 Interpretation		
		Normal	Impaired	Total	Normal	Impaired	Total
**Espiro App Pattern** **Interpretation**	Normal	74 (94.9%)	1 (2.5%)	75 (63.6%)	75 (98.7%)	0	75 (63.6%)
	Impaired	4 (5.1%)	39 (97.5%)	43 (36.4%)	1 (1.3%)	42 (100.0%)	43 (36.4%)
	Total	78	40	118	76	42	118

Values in parentheses show the percentage with respect to the total of the column (normal, impaired, and total).

**Table 2 diagnostics-14-00029-t002:** Validity of the Espiro app in pattern interpretation (normal and impaired).

	Pulmonologist 1	Pulmonologist 2
Sensitivity, %	97.5 (86.8–99.9)	100 (91.6–100)
Specificity, %	94.9 (87.4–98.6)	98.7 (92.9–99.9)
PPV, %	90.7 (78.9–96.2)	97.7 (85.7–99.7)
NPV, %	98.7 (91.4–99.8)	100 (93.9–100)
+LR	19.01 (7.31–49.45)	76 (10.84–532.61)
−LR	0.03 (0.00–0.18)	0
Accuracy, %	95.8 (90.4–98.6)	99.2 (95.4–99.9)

Values in parentheses are 95% confidence intervals. −LR: negative likelihood ratio; +LR: positive likelihood ratio; NPV: negative predictive value; PPV: positive predictive value.

**Table 3 diagnostics-14-00029-t003:** Cross-tabulation of spirometry pattern interpretation (normal, obstructive, spirometric restriction, mixed) by the Espiro app and each pulmonologist.

		Pulmonologist 1 Pattern Interpretation ^a^				Pulmonologist 2 Pattern Interpretation ^b^				
		Normal	Obstructive	Spirometric Restriction	Total	Normal	Obstructive	Spirometric Restriction	Mixed	Total
**Espiro App** **Pattern** **Interpretation**	Normal	74 (94.9%)	1 (4.0%)	0	75 (63.6%)	75 (98.7%)	0	0	0	75 (63.6%)
	Obstructive	1 (1.3%)	23 (92.0%)	1 (6.7%)	25 (21.2%)	0	11 (91.7%)	0	0	11 (9.3%)
	Spirometric restriction	3 (3.8%)	1 (4.0%)	14 (93.3%)	18 (15.3%)	1 (1.3%)	0	14 (100.0%)	3 (18.8%)	18 (15.3%)
	Mixed					0	1 (8.3%)	0	13 (81.3%)	14 (11.9%)
	Total	78	25	15	118	76	12	14	16	118

Values in parentheses show the percentage with respect to the total of the column (normal, obstructive, spirometric restriction, and total). ^a^ Kappa agreement (95% confidence interval) for the Espiro app and pulmonologist 1: 0.885 (0.803–0.967). The mixed pattern interpretation of the Espiro app to compare with pulmonologist 1 was included in the obstructive pattern because the pulmonologist did not specify whether the pattern was only obstructive or mixed. ^b^ Kappa agreement (95% confidence interval) for the Espiro app and pulmonologist 2: 0.923 (0.858–0.987).

**Table 4 diagnostics-14-00029-t004:** Values of the concordance coefficient kappa for pattern interpretation (normal or impaired) of the Espiro app and each of the observers (including the intra-rater reliability).

	Pulmonologist 1 (*n =* 118) ^a^	Pulmonologist 2 (*n =* 118) ^a^	Primary Care Physician 1 (*n =* 106) ^a^	Primary Care Physician 2 (*n =* 109) ^a^	Resident in Training Program 1 (*n =* 107) ^a^	Resident in Training Program 2 (*n =* 118) ^a^
**Espiro App**	0.907 (0.826–0.987)	0.982 (0.946–1.017)	0.980 (0.940–1.019)	1.000	0.978 (0.934–1.021)	1.000
**Pulmonologist 1** **(*n =* 46) ^b^**	0.953 (0.860–1.045)	0.888 (0.799–0.976)	0.918 (0.839–0.996)	0.919 (0.843–0.995)	0.885 (0.787–0.983)	0.907 (0.826–0.987)
**Pulmonologist 2** **(*n =* 49) ^b^**		1.000	0.959 (0.904–1.013)	1.000	0.978 (0.934–1.021)	0.982 (0.946–1.017)
**Primary Care Physician 1** **(*n =* 46) ^b^**			1.000	1.000	0.976 (0.928–1.023)	0.980 (0.940–1.019)
**Primary Care Physician 2** **(*n =* 45) ^b^**				1.000	0.976 (0.928–1.023)	1.000
**Resident in Training Program 1** **(*n =* 36) ^b^**					0.920 (0.774–1.071)	0.978 (0.934–1.021)
**Resident in Training Program 2** **(*n =* 49) ^b^**						1.000

Data are presented as kappa index (95% confidence interval). ^a^ Valid cases to evaluate the concordance with Espiro app interpretation. ^b^ Valid cases for intra-rater reliability.

**Table 5 diagnostics-14-00029-t005:** Values of the concordance coefficient kappa for severity interpretation of the Espiro app and each of the observers (including the intra-rater reliability).

	Pulmonologist 1 (*n =* 98) ^a^	Pulmonologist 2 (*n =* 105) ^a^	Primary Care Physician 1 (*n =* 106) ^a^	Primary Care Physician 2 (*n =* 107) ^a^	Resident in Training Program 1 (*n =* 105) ^a^	Resident in Training Program 2 (*n =* 102) ^a^
**Espiro App**	0.807 (0.675–0.938)	0.956 (0.895–1.016)	0.877 (0.792–0.961)	0.929 (0.862–0.995)	0.915 (0.836–0.993)	0.930 (0.853–1.006)
**Pulmonologist 1** **(*n =* 37) ^b^**	0.843 (0.643–1.042)	0.716 (0.523–0.908)	0.780 (0.646–0.913)	0.814 (0.678–0.949)	0.795 (0.659–0.930)	0.823 (0.691–0.954)
**Pulmonologist 2** **(*n =* 39) ^b^**		1.000	0.904 (0.813–0.994)	0.976 (0.930–1.021)	0.968 (0.905–1.030)	1.000
**Primary Care Physician 1** **(*n =* 46) ^b^**			0.963 (0.892–1.033)	0.943 (0.880–1.005)	0.861 (0.761–0.960)	0.845 (0.733–0.956)
**Primary Care Physician 2** **(*n =* 42) ^b^**				1.000	0.903 (0.812–0.993)	1.000
**Resident in Training Program 1** **(*n =* 36) ^b^**					0.784 (0.576–0.991)	0.901 (0.810–0.991)
**Resident in Training Program 2** **(*n =* 41) ^b^**						0.952 (0.861–1.042)

Data are presented as kappa index (95% confidence interval). The case numbers are different because the participants did not report the severity in all the spirometries evaluated. ^a^ Valid cases to evaluate the concordance with Espiro app interpretation. ^b^ Valid cases for intra-rater reliability.

## Data Availability

The data are readily available upon request.

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
