# Peer review of "Validity of the Espiro Mobile Application in the Interpretation of Spirometric Patterns: An App Accuracy Study"

_diagnostics, 2023, doi:10.3390/diagnostics14010029_

Round 1

Reviewer 1 Report

Comments and Suggestions for Authors

1. The main question addressed by the research is to assess the interpretation accuracy of the Espiro App, using the pulmonologist interpretation as the gold standard.

2. The topic is original and it adds some new information.

3. The comparison of interpretation of App, pulmonologist and primary care physicians is interesting and new.

4. Authors should consider to test the App as a system which may evaluate the technical acceptabilitu of blows, as spirometer is a test which primary physicians should be able to perform in their office.

5. As Authors mention primary care plays an important role in the early diagnosis of respiratory diseases and spirometry is the most widely used PFT in primary care and is considered the “gold  standard” to evaluate the most common causes of pulmonary disease. For this reason, Authors should discuss that,  In addition to systems that help in the interpretation of the results, it is essential to have systems that evaluate the technical acceptability of  blows that have been performed.

Comments on the Quality of English Language

Minor editing of English language required

Author Response

Point by point responses to Reviewer 1

Thank you for your suggestions, we have incorporated them into the manuscript, and you will find them highlighted in yellow colour.

The text that must be delated is highlighted in red.

The main question addressed by the research is to assess the interpretation accuracy of the Espiro App, using the pulmonologist interpretation as the gold standard.

The topic is original and it adds some new information.

The comparison of interpretation of App, pulmonologist and primary care physicians is interesting and new.

We appreciate your positive response to our manuscript titled " Validity of the Espiro Mobile Application in the Interpretation of Spirometric Patterns: An App Accuracy Study." Your dedicated time in providing valuable feedback is sincerely acknowledged, and we express our gratitude for your insightful comments on our article.

Authors should consider to test the App as a system which may evaluate the technical acceptability of blows, as spirometer is a test which primary physicians should be able to perform in their office.

Suggestions provided have been carefully considered, and we have implemented the necessary changes in the revised version of the manuscript. Specifically, we introduce this paragraph in 2.Materials and methods section, subheading 2.3. Espiro APP: “ The App don’t evaluate the technical acceptability of the spirometric technique because it was designed to interpretate the spirometric patterns. The evaluation of the quality of the technique needs to be made by the technician that perform the test or the physician that evaluate it.” Thank you for your comment.

As Authors mention primary care plays an important role in the early diagnosis of respiratory diseases and spirometry is the most widely used PFT in primary care and is considered the “gold standard” to evaluate the most common causes of pulmonary disease. For this reason, Authors should discuss that, In addition to systems that help in the interpretation of the results, it is essential to have systems that evaluate the technical acceptability of blows that have been performed.

Thank you for your comment, we commented this in the section 4.2. Implications for clinical practice and research (lines 504-509), also we added a new paragraph to clarify more this topic (lines 510-514).

Comments on the Quality of English Language

Minor editing of English language required

Thank you for all your contributions. Our article has been reviewed by a native English speaker, one of the authors (Iñaki Voelcker-Sala). Likewise, if deemed necessary, we are open to a review by the editorial services.

Reviewer 2 Report

Comments and Suggestions for Authors

Title of the article: Validity of the Espiro Mobile Application in the Interpretation of Spirometric Patterns: An App Accuracy Study

My comments:

                In general, this is an interesting research. The authors aim to interpretation accuracy of the Espiro App, using pulmonologist interpretations as the gold standard. However, there are some points should be improved.           

1. Abstract

            Adequate and nicely written.

2. Introduction

             Adequate and nicely written

3. Materials and Methods

            3.1.         Restrictive should be changed to spirometric restriction. Because this restrictive was not confirmed by total lung measurement.

                3.2.         Please provide the reference used for the severity classification of airflow limitation or restriction, e.g.: ATS/ERS 2005.

                3.3          In 2021, the definition of bronchodilator response (BDR) was change to 10% change relative to predicted value (GLI 2012 predictive value). However, when you classified a positive of BDR with an improvement in the FEV1 or the FVC 12% and 0.2 L, the reference should be provided. Additionally, the concordance for the BDR test was not shown in the results section. Thus, I recommend deleting the BDR paragraph in the methods section.

.

4. Results

            4.1. Clearly present

5. Discussion

            Results of the present investigation have been written adequately and discussion is supported with scientific factual reasoning. However, the discussion about the BDR should be deleted. Because there is no results of the concordance for the BDR test in the results section.

5. Conclusion

            Adequate and nicely written.

7. References

            All the cited references should be changed to the journal’s format.      

Author Response

Point by point responses to Reviewer 2

Thank you for your suggestions, we have incorporated them into the manuscript, and you will find them highlighted in blue colour.

The text that must be delated is highlighted in red.

In general, this is an interesting research. The authors aim to interpretation accuracy of the Espiro App, using pulmonologist interpretations as the gold standard. However, there are some points should be improved.

We appreciate your positive response to our manuscript titled " Validity of the Espiro Mobile Application in the Interpretation of Spirometric Patterns: An App Accuracy Study." Your dedicated time in providing valuable feedback is sincerely acknowledged, and we express our gratitude for your insightful comments on our article.

Suggestions provided have been carefully considered, and we have implemented the necessary changes in the revised version of the manuscript, as outlined in the attached "point-by-point" document.

  1. Abstract: Adequate and nicely written.
  2. Introduction: Adequate and nicely written.

Thank you for your kind words on the abstract and introduction “Adequate and nicely written”. We appreciate your positive feedback.

  1. Materials and Methods

3.1. Restrictive should be changed to spirometric restriction. Because this restrictive was not confirmed by total lung measurement.

Thank you for your comment. We have addressed your suggestion and replaced “restrictive” with “spirometric restriction”. This modification ensures clarity, as the use of 'spirometric restriction' is more appropriate.

3.2. Please provide the reference used for the severity classification of airflow limitation or restriction, e.g.: ATS/ERS 2005.

Thank you for your suggestion. We have incorporated the requested reference for the severity classification of airflow limitation or restriction, specifically citing SEPAR criteria in the text.

3.3. In 2021, the definition of bronchodilator response (BDR) was change to 10% change relative to predicted value (GLI 2012 predictive value). However, when you classified a positive of BDR with an improvement in the FEV1 or the FVC ≥ 12% and ≥ 0.2 L, the reference should be provided.

Thank you for bringing this to our attention. We have adjusted our classification criteria in accordance with SEPAR criteria of BDR. In particular, we have incorporated the reference into our paper.

Additionally, the concordance for the BDR test was not shown in the results section. Thus, I recommend deleting the BDR paragraph in the methods section.

Thank you for your contribution. In section “2.3. Spiro App, comprehensive details interpreting our application and the criteria used, including the BDR, are presented to the reader. Therefore, despite your suggestion, we consider it pertinent to retain the last paragraph that refers to the BDR. As you rightly highlight, concordance data for the BDR are not shown in the results section. As we have indicated at the end of the section, the reason was “The concordance for the bronchodilator test could not be calculated due to insufficient interpretations performed by the different observers”. We intend to address and discuss this limitation in the discussion section.

We introduced a new figure in the Material and Methods section, under subheading 2.3, displaying screenshots of the application to illustrate its functionality and performance.

  1. Results: Clearly present

Thank you for your comment.

  1. Discussion

Results of the present investigation have been written adequately and discussion is supported with scientific factual reasoning.

Thank you for your comment.

However, the discussion about the BDR should be deleted. Because there is no results of the concordance for the BDR test in the results section.

As previously mentioned, the content of this discussion paragraph has been relocated to the “Limitations and Strengths” section, where it is now addressed as one of the limitations of our study.

  1. Conclusion: Adequate and nicely written.

Thank you for your comment.

  1. References: All the cited references should be changed to the journal’s format.

Thank you for your advice. We have adapted the references to the journal's standards.

Round 2

Reviewer 2 Report

Comments and Suggestions for Authors

All of my comments have been addressed by authors. This manuscript can be accepted as a current form.